# FOR INTERPOLATING KERNEL MACHINES, MINIMIZING THE NORM OF THE ERM SOLUTION MINIMIZES STABILITY

## ABSTRACT

We study the average $\text{CV}_{loo}$ stability of kernel ridge-less regression and derive corresponding risk bounds. We show that the interpolating solution with minimum norm minimizes a bound on $\text{CV}_{loo}$ stability, which in turn is controlled by the condition number of the empirical kernel matrix. The latter can be characterized in the asymptotic regime where both the dimension and cardinality of the data go to infinity. Under the assumption of random kernel matrices, the corresponding test error should be expected to follow a double descent curve.

## 1 INTRODUCTION

Statistical learning theory studies the learning properties of machine learning algorithms, and more fundamentally, the conditions under which learning from finite data is possible. In this context, classical learning theory focuses on the size of the hypothesis space in terms of different complexity measures, such as combinatorial dimensions, covering numbers and Rademacher/Gaussian complexities (Shalev-Shwartz & Ben-David, 2014; Boucheron et al., 2005). Another more recent approach is based on defining suitable notions of stability with respect to perturbation of the data (Bousquet & Elisseeff, 2001; Kutin & Niyogi, 2002). In this view, the continuity of the process that maps data to estimators is crucial, rather than the complexity of the hypothesis space. Different notions of stability can be considered, depending on the data perturbation and metric considered (Kutin & Niyogi, 2002). Interestingly, the stability and complexity approaches to characterizing the *learnability* of problems are not at odds with each other, and can be shown to be equivalent as shown in Poggio et al. (2004) and Shalev-Shwartz et al. (2010).

In modern machine learning overparameterized models, with a larger number of parameters than the size of the training data, have become common. The ability of these models to generalize is well explained by classical statistical learning theory as long as some form of regularization is used in the training process (Bühlmann & Van De Geer, 2011; Steinwart & Christmann, 2008). However, it was recently shown - first for deep networks (Zhang et al., 2017), and more recently for kernel methods (Belkin et al., 2019) - that learning is possible in the absence of regularization, i.e., when perfectly fitting/interpolating the data. Much recent work in statistical learning theory has tried to find theoretical ground for this empirical finding. Since learning using models that interpolate is not exclusive to deep neural networks, we study generalization in the presence of interpolation in the case of kernel methods. We study both linear and kernel least squares problems in this paper.

**Our Contributions:**

- We characterize the generalization properties of interpolating solutions for linear and kernel least squares problems using a stability approach. While the (uniform) stability properties of regularized kernel methods are well known (Bousquet & Elisseeff, 2001), we study interpolating solutions of the unregularized ("ridgeless") regression problems.

- We obtain an upper bound on the stability of interpolating solutions, and show that this upper bound is minimized by the minimum norm interpolating solution. This also means that among all interpolating solutions, the minimum norm solution has the best test error. In

particular, the same conclusion is also true for gradient descent, since it converges to the minimum norm solution in the setting we consider, see e.g. Rosasco & Villa (2015).

- Our stability bounds show that the average stability of the minimum norm solution is controlled by the condition number of the empirical kernel matrix. It is well known that the numerical stability of the least squares solution is governed by the condition number of the associated kernel matrix (see the discussion of why overparametrization is "good" in Poggio et al. (2019)). Our results show that the condition number also controls stability (and hence, test error) in a statistical sense.

**Organization:** In section 2, we introduce basic ideas in statistical learning and empirical risk minimization, as well as the notation used in the rest of the paper. In section 3, we briefly recall some definitions of stability. In section 4, we study the stability of interpolating solutions to kernel least squares and show that the minimum norm solutions minimize an upper bound on the stability. In section 5 we discuss our results in the context of recent work on high dimensional regression. We conclude in section 6.

## 2    STATISTICAL LEARNING AND EMPIRICAL RISK MINIMIZATION

We begin by recalling the basic ideas in statistical learning theory. In this setting, $X$ is the space of features, $Y$ is the space of targets or labels, and there is an unknown probability distribution $\mu$ on the product space $Z = X \times Y$. In the following, we consider $X = \mathbb{R}^d$ and $Y = \mathbb{R}$. The distribution $\mu$ is fixed but unknown, and we are given a training set $S$ consisting of $n$ samples (thus $|S| = n$) drawn i.i.d. from the probability distribution on $Z^n$, $S = (z_i)_{i=1}^n = (\mathbf{x}_i, y_i)_{i=1}^n$. Intuitively, the goal of supervised learning is to use the training set $S$ to "learn" a function $f_S$ that evaluated at a new value $\mathbf{x}_{new}$ should predict the associated value of $y_{new}$, i.e. $y_{new} \approx f_S(\mathbf{x}_{new})$.

The loss is a function $V : \mathcal{F} \times Z \to [0, \infty)$, where $\mathcal{F}$ is the space of measurable functions from $X$ to $Y$, that measures how well a function performs on a data point. We define a hypothesis space $\mathcal{H} \subseteq \mathcal{F}$ where algorithms search for solutions. With the above notation, the *expected risk* of $f$ is defined as $I[f] = \mathbb{E}_z V(f, z)$ which is the expected loss on a new sample drawn according to the data distribution $\mu$. In this setting, statistical learning can be seen as the problem of finding an approximate minimizer of the expected risk given a training set $S$. A classical approach to derive an approximate solution is empirical risk minimization (ERM) where we minimize the empirical risk $I_S[f] = \frac{1}{n} \sum_{i=1}^n V(f, z_i)$.

A natural error measure for our ERM solution $f_S$ is the expected excess risk $\mathbb{E}_S[I[f_S] - \min_{f \in \mathcal{H}} I[f]]$. Another common error measure is the expected generalization error/gap given by $\mathbb{E}_S[I[f_S] - I_S[f_S]]$. These two error measures are closely related since, the expected excess risk is easily bounded by the expected generalization error (see Lemma 5).

### 2.1    KERNEL LEAST SQUARES AND MINIMUM NORM SOLUTION

The focus in this paper is on the kernel least squares problem. We assume the loss function $V$ is the square loss, that is, $V(f, z) = (y - f(\mathbf{x}))^2$. The hypothesis space is assumed to be a reproducing kernel Hilbert space, defined by a positive definite kernel $K : X \times X \to \mathbb{R}$ or an associated feature map $\Phi : X \to \mathcal{H}$, such that $K(\mathbf{x}, \mathbf{x}') = \langle \Phi(\mathbf{x}), \Phi(\mathbf{x}') \rangle_{\mathcal{H}}$ for all $\mathbf{x}, \mathbf{x}' \in X$, where $\langle \cdot, \cdot \rangle_{\mathcal{H}}$ is the inner product in $\mathcal{H}$. In this setting, functions are linearly parameterized, that is there exists $w \in \mathcal{H}$ such that $f(\mathbf{x}) = \langle w, \Phi(\mathbf{x}) \rangle_{\mathcal{H}}$ for all $x \in X$.

The ERM problem typically has multiple solutions, one of which is the minimum norm solution:

$$f_S^\dagger = \arg\min_{f \in \mathcal{M}} \|f\|_{\mathcal{H}}, \qquad \mathcal{M} = \arg\min_{f \in \mathcal{H}} \frac{1}{n} \sum_{i=1}^n (f(\mathbf{x}_i) - y_i)^2. \tag{1}$$

Here $\|\cdot\|_{\mathcal{H}}$ is the norm on $\mathcal{H}$ induced by the inner product. The minimum norm solution can be shown to be unique and satisfy a representer theorem, that is for all $\mathbf{x} \in X$:

$$f_S^\dagger(\mathbf{x}) = \sum_{i=1}^n K(\mathbf{x}, \mathbf{x}_i) \mathbf{c}_S[i], \qquad \mathbf{c}_S = \mathbf{K}^\dagger \mathbf{y} \tag{2}$$

where $\mathbf{c}_S = (\mathbf{c}_S[1], \ldots, \mathbf{c}_S[n]), \mathbf{y} = (y_1 \ldots y_n) \in \mathbb{R}^n$, $\mathbf{K}$ is the $n$ by $n$ matrix with entries $\mathbf{K}_{ij} = K(\mathbf{x}_i, \mathbf{x}_j), i, j = 1, \ldots, n$, and $\mathbf{K}^\dagger$ is the Moore-Penrose pseudoinverse of $\mathbf{K}$. If we assume $n \leq d$ and that we have $n$ linearly independent data features, that is the rank of $\mathbf{X}$ is $n$, then it is possible to show that for many kernels one can replace $\mathbf{K}^\dagger$ by $\mathbf{K}^{-1}$ (see Remark 2). Note that invertibility is necessary and sufficient for interpolation. That is, if $\mathbf{K}$ is invertible, $f_S^\dagger(\mathbf{x}_i) = y_i$ for all $i = 1, \ldots, n$, in which case the training error in (1) is zero.

**Remark 1 (Pseudoinverse for underdetermined linear systems)** *A simple yet relevant example are linear functions $f(\mathbf{x}) = \mathbf{w}^\top \mathbf{x}$, that correspond to $\mathcal{H} = \mathbb{R}^d$ and $\Phi$ the identity map. If the rank of $\mathbf{X} \in \mathbb{R}^{d \times n}$ is $n$, then any interpolating solution $\mathbf{w}_S$ satisfies $\mathbf{w}_S^\top \mathbf{x}_i = y_i$ for all $i = 1, \ldots, n$, and the minimum norm solution, also called Moore-Penrose solution, is given by $(\mathbf{w}_S^\dagger)^\top = \mathbf{y}^\top \mathbf{X}^\dagger$ where the pseudoinverse $\mathbf{X}^\dagger$ takes the form $\mathbf{X}^\dagger = \mathbf{X}^\top (\mathbf{X}\mathbf{X}^\top)^{-1}$.*

**Remark 2 (Invertibility of translation invariant kernels)** *Translation invariant kernels are a family of kernel functions given by $K(\mathbf{x}_1, \mathbf{x}_2) = k(\mathbf{x}_1 - \mathbf{x}_2)$ where $k$ is an even function on $\mathbb{R}^d$. Translation invariant kernels are Mercer kernels (positive semidefinite) if the Fourier transform of $k(\cdot)$ is non-negative. For Radial Basis Function kernels $(K(\mathbf{x}_1, \mathbf{x}_2) = k(\|\mathbf{x}_1 - \mathbf{x}_2\|))$ we have the additional property due to Theorem 2.3 of Micchelli (1986) that for distinct points $\mathbf{x}_1, \mathbf{x}_2, \ldots, \mathbf{x}_n \in \mathbb{R}^d$ the kernel matrix $\mathbf{K}$ is non-singular and thus invertible.*

The above discussion is directly related to regularization approaches.

**Remark 3 (Stability and Tikhonov regularization)** *Tikhonov regularization is used to prevent potential unstable behaviors. In the above setting, it corresponds to replacing Problem (1) by $\min_{f \in \mathcal{H}} \frac{1}{n} \sum_{i=1}^n (f(\mathbf{x}_i) - y_i)^2 + \lambda \|f\|_{\mathcal{H}}^2$ where the corresponding unique solution is given by $f_S^\lambda(\mathbf{x}) = \sum_{i=1}^n K(\mathbf{x}, \mathbf{x}_i)\mathbf{c}[i], \qquad \mathbf{c} = (\mathbf{K} + \lambda \mathbf{I}_n)^{-1}\mathbf{y}$. In contrast to ERM solutions, the above approach prevents interpolation. The properties of the corresponding estimator are well known. In this paper, we complement these results focusing on the case $\lambda \to 0$.*

Finally, we end by recalling the connection between minimum norm and the gradient descent.

**Remark 4 (Minimum norm and gradient descent)** *In our setting, it is well known that both batch and stochastic gradient iterations converge exactly to the minimum norm solution when multiple solutions exist, see e.g. Rosasco & Villa (2015). Thus, a study of the properties of the minimum norm solution explains the properties of the solution to which gradient descent converges. In particular, when ERM has multiple interpolating solutions, gradient descent converges to a solution that minimizes a bound on stability, as we show in this paper.*

## 3 ERROR BOUNDS VIA STABILITY

In this section, we recall basic results relating the learning and stability properties of Empirical Risk Minimization (ERM). Throughout the paper, we assume that ERM achieves a minimum, albeit the extension to almost minimizer is possible (Mukherjee et al., 2006) and important for exponential-type loss functions (Poggio, 2020). We do not assume the expected risk to achieve a minimum. Since we will be considering leave-one-out stability in this section, we look at solutions to ERM over the complete training set $S = \{z_1, z_2, \ldots, z_n\}$ and the leave one out training set $S_i = \{z_1, z_2, \ldots, z_{i-1}, z_{i+1}, \ldots, z_n\}$

The excess risk of ERM can be easily related to its stability properties. Here, we follow the definition laid out in Mukherjee et al. (2006) and say that an algorithm is Cross-Validation leave-one-out (CV$_{loo}$) stable in expectation, if there exists $\beta_{CV} > 0$ such that for all $i = 1, \ldots, n$,

$$\mathbb{E}_S[V(f_{S_i}, z_i) - V(f_S, z_i)] \leq \beta_{CV}. \tag{3}$$

This definition is justified by the following result that bounds the excess risk of a learning algorithm by its average CV$_{loo}$ stability (Shalev-Shwartz et al., 2010; Mukherjee et al., 2006).

**Lemma 5 (Excess Risk & CV$_{loo}$ Stability)** *For all $i = 1, \ldots, n$,*

$$\mathbb{E}_S[I[f_{S_i}] - \inf_{f \in \mathcal{H}} I[f]] \leq \mathbb{E}_S[V(f_{S_i}, z_i) - V(f_S, z_i)]. \tag{4}$$

**Remark 6 (Connection to uniform stability and other notions of stability)** *Uniform stability, introduced by Bousquet & Elisseeff (2001), corresponds in our notation to the assumption that there exists $\beta_u > 0$ such that for all $i = 1, \ldots, n$, $\sup_{z \in Z} |V(f_{S_i}, z) - V(f_S, z)| \leq \beta_u$. Clearly this is a strong notion implying most other definitions of stability. We note that there are number of different notions of stability. We refer the interested reader to Kutin & Niyogi (2002), Mukherjee et al. (2006).*

We recall the proof of Lemma 5 in Appendix A.2 due to lack of space. In Appendix A, we also discuss other definitions of stability and their connections to concepts in statistical learning theory like generalization and learnability.

## 4 $\mathrm{CV}_{loo}$ STABILITY OF KERNEL LEAST SQUARES

In this section we analyze the expected $\mathrm{CV}_{loo}$ stability of interpolating solutions to the kernel least squares problem, and obtain an upper bound on their stability. We show that this upper bound on the expected $\mathrm{CV}_{loo}$ stability is smallest for the minimum norm interpolating solution (1) when compared to other interpolating solutions to the kernel least squares problem.

We have a dataset $S = \{(\mathbf{x}_i, y_i)\}_{i=1}^n$ and we want to find a mapping $f \in \mathcal{H}$, that minimizes the empirical least squares risk. Here $\mathcal{H}$ is a reproducing kernel hilbert space (RKHS) defined by a positive definite kernel $K : X \times X \to \mathbb{R}$. All interpolating solutions are of the form $\hat{f}_S(\cdot) = \sum_{j=1}^n \hat{\mathbf{c}}_S[j] K(\mathbf{x}_j, \cdot)$, where $\hat{\mathbf{c}}_S = \mathbf{K}^\dagger \mathbf{y} + (\mathbf{I} - \mathbf{K}^\dagger \mathbf{K})\mathbf{v}$. Similarly, all interpolating solutions on the leave one out dataset $S_i$ can be written as $\hat{f}_{S_i}(\cdot) = \sum_{j=1, j \neq i}^n \hat{\mathbf{c}}_{S_i}[j] K(\mathbf{x}_j, \cdot)$, where $\hat{\mathbf{c}}_{S_i} = \mathbf{K}_{S_i}^\dagger \mathbf{y}_i + (\mathbf{I} - \mathbf{K}_{S_i}^\dagger \mathbf{K}_{S_i})\mathbf{v}_i$. Here $\mathbf{K}, \mathbf{K}_{S_i}$ are the empirical kernel matrices on the original and leave one out datasets respectively. We note that when $\mathbf{v} = \mathbf{0}$ and $\mathbf{v}_i = \mathbf{0}$, we obtain the minimum norm interpolating solutions on the datasets $S$ and $S_i$.

**Theorem 7 (Main Theorem)** *Consider the kernel least squares problem with a bounded kernel and bounded outputs $y$, that is there exist $\kappa, M > 0$ such that*

$$K(\mathbf{x}, \mathbf{x}') \leq \kappa^2, \qquad |y| \leq M, \tag{5}$$

*almost surely. Then for any interpolating solutions $\hat{f}_{S_i}, \hat{f}_S$,*

$$\mathbb{E}_S[V(\hat{f}_{S_i}, z_i) - V(\hat{f}_S, z_i)] \leq \beta_{CV}(\mathbf{K}^\dagger, \mathbf{y}, \mathbf{v}, \mathbf{v}_i) \tag{6}$$

*This bound $\beta_{CV}$ is minimized when $\mathbf{v} = \mathbf{v}_i = \mathbf{0}$, which corresponds to the minimum norm interpolating solutions $f_S^\dagger, f_{S_i}^\dagger$. For the minimum norm solutions we have $\beta_{CV} = C_1 \beta_1 + C_2 \beta_2$, where $\beta_1 = \mathbb{E}_S \left[ \|\mathbf{K}^{\frac{1}{2}}\|_{op} \|\mathbf{K}^\dagger\|_{op} \times cond(\mathbf{K}) \times \|\mathbf{y}\| \right]$ and, $\beta_2 = \mathbb{E}_S \left[ \|\mathbf{K}^{\frac{1}{2}}\|_{op}^2 \|\mathbf{K}^\dagger\|_{op}^2 \times (cond(\mathbf{K}))^2 \times \|\mathbf{y}\|^2 \right]$, and $C_1, C_2$ are absolute constants that do not depend on either $d$ or $n$.*

In the above theorem $\|\mathbf{K}\|_{op}$ refers to the operator norm of the kernel matrix $\mathbf{K}$, $\|\mathbf{y}\|$ refers to the standard $\ell_2$ norm for $\mathbf{y} \in \mathbb{R}^n$, and $cond(\mathbf{K})$ is the condition number of the matrix $\mathbf{K}$.

We can combine the above result with Lemma 5 to obtain the following bound on excess risk for minimum norm interpolating solutions to the kernel least squares problem:

**Corollary 8** *The excess risk of the minimum norm interpolating kernel least squares solution can be bounded as:*

$$\mathbb{E}_S \left[ I[f_{S_i}^\dagger] - \inf_{f \in \mathcal{H}} I[f] \right] \leq C_1 \beta_1 + C_2 \beta_2$$

*where $\beta_1, \beta_2$ are as defined previously.*

**Remark 9 (Underdetermined Linear Regression)** *In the case of underdetermined linear regression, ie, linear regression where the dimensionality is larger than the number of samples in the training set, we can prove a version of Theorem 7 with $\beta_1 = \mathbb{E}_S \left[ \|\mathbf{X}^\dagger\|_{op} \|\mathbf{y}\| \right]$ and $\beta_2 = \mathbb{E}_S \left[ \|\mathbf{X}^\dagger\|_{op}^2 \|\mathbf{y}\|^2 \right]$. Due to space constraints, we present the proof of the results in the linear regression case in Appendix B.*

### 4.1 Key lemmas

In order to prove Theorem 7 we make use of the following lemmas to bound the $\text{CV}_{loo}$ stability using the norms and the difference of the solutions.

**Lemma 10** *Under assumption (5), for all $i = 1\ldots,n$, it holds that*

$$\mathbb{E}_S[V(\hat{f}_{S_i}, z_i) - V(\hat{f}_S, z_i)] \leq \mathbb{E}_S\left[\left(2M + \kappa\left(\left\|\hat{f}_S\right\|_{\mathcal{H}} + \left\|\hat{f}_{S_i}\right\|_{\mathcal{H}}\right)\right) \times \kappa\left\|\hat{f}_S - \hat{f}_{S_i}\right\|_{\mathcal{H}}\right]$$

**Proof** We begin, recalling that the square loss is locally Lipschitz, that is for all $y, a, a' \in \mathbb{R}$, with

$$|(y - a)^2 - (y - a')^2| \leq (2|y| + |a| + |a'|))|a - a'|.$$

If we apply this result to $f, f'$ in a RKHS $\mathcal{H}$,

$$|(y - f(\mathbf{x}))^2 - (y - f'(\mathbf{x}))^2| \leq \kappa(2M + \kappa(\|f\|_{\mathcal{H}} + \|f'\|_{\mathcal{H}}))\|f - f'\|_{\mathcal{H}}.$$

using the basic properties of a RKHS that for all $f \in \mathcal{H}$

$$|f(\mathbf{x})| \leq \|f\|_{\infty} = \sup_{\mathbf{x}}|f(\mathbf{x})| = \sup_{\mathbf{x}}|\langle f, K_{\mathbf{x}}\rangle_{\mathcal{H}}| \leq \kappa\|f\|_{\mathcal{H}} \tag{7}$$

In particular, we can plug $\hat{f}_{S_i}$ and $\hat{f}_S$ into the above inequality, and the almost positivity of ERM (Mukherjee et al., 2006) will allow us to drop the absolute value on the left hand side. Finally the desired result follows by taking the expectation over $S$. ∎

Now that we have bounded the $\text{CV}_{loo}$ stability using the norms and the difference of the solutions, we can find a bound on the difference between the solutions to the kernel least squares problem. This is our main stability estimate.

**Lemma 11** *Let $\hat{f}_S, \hat{f}_{S_i}$ be any interpolating kernel least squares solutions on the full and leave one out datasets (as defined at the top of this section), then $\left\|\hat{f}_S - \hat{f}_{S_i}\right\|_{\mathcal{H}} \leq B_{CV}(\mathbf{K}^{\dagger}, \mathbf{y}, \mathbf{v}, \mathbf{v}_i)$, and $B_{CV}$ is minimized when $\mathbf{v} = \mathbf{v}_i = \mathbf{0}$, which corresponds to the minimum norm interpolating solutions $f_S^{\dagger}, f_{S_i}^{\dagger}$.*

*Also for some absolute constant $C$,*

$$\left\|f_S^{\dagger} - f_{S_i}^{\dagger}\right\|_{\mathcal{H}} \leq C \times \left\|\mathbf{K}^{\frac{1}{2}}\right\|_{op} \left\|\mathbf{K}^{\dagger}\right\|_{op} \times cond(\mathbf{K}) \times \|\mathbf{y}\| \tag{8}$$

Since the minimum norm interpolating solutions minimize both $\left\|\hat{f}_S\right\|_{\mathcal{H}} + \left\|\hat{f}_{S_i}\right\|_{\mathcal{H}}$ and $\left\|\hat{f}_S - \hat{f}_{S_i}\right\|_{\mathcal{H}}$ (from lemmas 10, 11), we can put them together to prove theorem 7. In the following section we provide the proof of Lemma 11.

**Remark 12 (Zero training loss)** *In Lemma 10 we use the locally Lipschitz property of the squared loss function to bound the leave one out stability in terms of the difference between the norms of the solutions. Under interpolating conditions, if we set the term $V(\hat{f}_S, z_i) = 0$, the leave one out stability reduces to $\mathbb{E}_S\left[V(\hat{f}_{S_i}, z_i) - V(\hat{f}_S, z_i)\right] = \mathbb{E}_S\left[V(\hat{f}_{S_i}, z_i)\right] = \mathbb{E}_S[(\hat{f}_{S_i}(\mathbf{x}_i) - y_i)^2] = \mathbb{E}_S[(\hat{f}_{S_i}(\mathbf{x}_i) - \hat{f}_S(\mathbf{x}_i))^2] = \mathbb{E}_S[\langle\hat{f}_{S_i}(\cdot) - \hat{f}_S(\cdot), K_{\mathbf{x}_i}(\cdot)\rangle^2] \leq \mathbb{E}_S\left[||\hat{f}_S - \hat{f}_{S_i}||_{\mathcal{H}}^2 \times \kappa^2\right]$. We can plug in the bound from Lemma 11 to obtain similar qualitative and quantitative (up to constant factors) results as in Theorem 7.*

**Simulation:** In order to illustrate that the minimum norm interpolating solution is the best performing interpolating solution we ran a simple experiment on a linear regression problem. We synthetically generated data from a linear model $\mathbf{y} = \mathbf{w}^{\top}\mathbf{X}$, where $\mathbf{X} \in \mathbb{R}^{d \times n}$ was i.i.d $\mathcal{N}(0, 1)$. The dimension of the data was $d = 1000$ and there were $n = 200$ samples in the training dataset. A held out dataset of 50 samples was used to compute the test mean squared error (MSE). Interpolating solutions were computed as $\hat{\mathbf{w}}^{\top} = \mathbf{y}^{\top}\mathbf{X}^{\dagger} + \mathbf{v}^{\top}(\mathbf{I} - \mathbf{X}\mathbf{X}^{\dagger})$ and the norm of $\mathbf{v}$ was varied to obtain the plot. The results are shown in Figure 1, where we can see that the training loss is 0 for all interpolants, but test MSE increases as $||\mathbf{v}||$ increases, with $(\mathbf{w}^{\dagger})^{\top} = \mathbf{y}^{\top}\mathbf{X}^{\dagger}$ having the best performance. The figure reports results averaged over 100 trials.

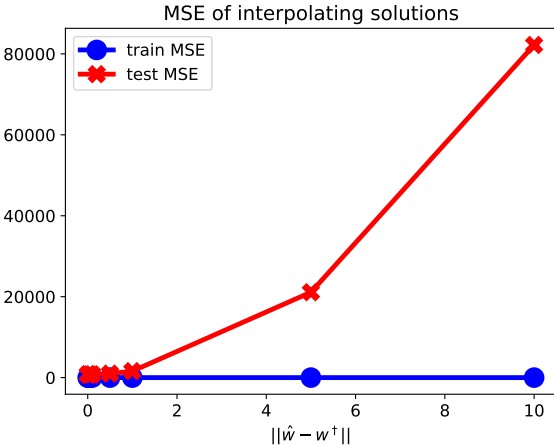

Figure 1: Plot of train and test mean squared error (MSE) vs distance between an interpolating solution $\hat{\mathbf{w}}$ and the minimum norm interpolant $\mathbf{w}^\dagger$ of a linear regression problem. Data was synthetically generated as $\mathbf{y} = \mathbf{w}^\top \mathbf{X}$, where $\mathbf{X} \in \mathbb{R}^{d \times n}$ with i.i.d $\mathcal{N}(0, 1)$ entries and $d = 1000, n = 200$. Other interpolating solutions were computed as $\hat{\mathbf{w}} = \mathbf{y}^\top \mathbf{X}^\dagger + \mathbf{v}^\top (\mathbf{I} - \mathbf{X}\mathbf{X}^\dagger)$ and the norm of $\mathbf{v}$ was varied to obtain the plot. Train MSE is 0 for all interpolants, but test MSE increases as $||\mathbf{v}||$ increases, with $\mathbf{w}^\dagger$ having the best performance. This plot represents results averaged over 100 trials.

## 4.2 PROOF OF LEMMA 11

We can write any interpolating solution to the kernel regression problem as $\hat{f}_S(\mathbf{x}) = \sum_{i=1}^n \hat{\mathbf{c}}_S[i] K(\mathbf{x}_i, \mathbf{x})$ where $\hat{\mathbf{c}}_S = \mathbf{K}^\dagger \mathbf{y} + (\mathbf{I} - \mathbf{K}^\dagger \mathbf{K})\mathbf{v}$, and $\mathbf{K} \in \mathbb{R}^{n \times n}$ is the kernel matrix $K$ on $S$ and $\mathbf{v}$ is any vector in $\mathbb{R}^n$. i.e. $\mathbf{K}_{ij} = K(\mathbf{x}_i, \mathbf{x}_j)$, and $\mathbf{y} \in \mathbb{R}^n$ is the vector $\mathbf{y} = [y_1 \ldots y_n]^\top$.

Similarly, the coefficient vector for the corresponding interpolating solution to the problem over the leave one out dataset $S_i$ is $\hat{\mathbf{c}}_{S_i} = (\mathbf{K}_{S_i})^\dagger \mathbf{y}_i + (\mathbf{I} - (\mathbf{K}_{S_i})^\dagger \mathbf{K}_{S_i})\mathbf{v}_i$. Where $\mathbf{y}_i = [y_1, \ldots, 0, \ldots y_n]^\top$ and $\mathbf{K}_{S_i}$ is the kernel matrix $\mathbf{K}$ with the $i^{\text{th}}$ row and column set to zero, which is the kernel matrix for the leave one out training set.

We define $\mathbf{a} = [-K(\mathbf{x}_1, \mathbf{x}_i), \ldots, -K(\mathbf{x}_n, \mathbf{x}_i)]^\top \in \mathbb{R}^n$ and $\mathbf{b} \in \mathbb{R}^n$ as a one-hot column vector with all zeros apart from the $i^{\text{th}}$ component which is 1. Let $\mathbf{a}_* = \mathbf{a} + K(\mathbf{x}_i, \mathbf{x}_i)\mathbf{b}$. Then, we have:

$$\mathbf{K}_* = \mathbf{K} + \mathbf{b}\mathbf{a}_*^\top$$
$$\mathbf{K}_{S_i} = \mathbf{K}_* + \mathbf{a}\mathbf{b}^\top \tag{9}$$

That is, we can write $\mathbf{K}_{S_i}$ as a rank-2 update to $\mathbf{K}$. This can be verified by simple algebra, and using the fact that $K$ is a symmetric kernel. Now we are interested in bounding $||\hat{f}_S - \hat{f}_{S_i}||_{\mathcal{H}}$. For a function $h(\cdot) = \sum_{i=1}^m p_i K(\mathbf{x}_i, \cdot) \in \mathcal{H}$ we have $||h||_{\mathcal{H}} = \sqrt{\mathbf{p}^\top \mathbf{K}\mathbf{p}} = ||\mathbf{K}^{\frac{1}{2}}\mathbf{p}||$. So we have:

$$
\begin{aligned}
||\hat{f}_S - \hat{f}_{S_i}||_{\mathcal{H}} &= ||\mathbf{K}^{\frac{1}{2}}(\hat{\mathbf{c}}_S - \hat{\mathbf{c}}_{S_i})|| \\
&= ||\mathbf{K}^{\frac{1}{2}}(\mathbf{K}^\dagger \mathbf{y} + (\mathbf{I} - \mathbf{K}^\dagger \mathbf{K})\mathbf{v} - (\mathbf{K}_{S_i})^\dagger \mathbf{y}_i - (\mathbf{I} - (\mathbf{K}_{S_i})^\dagger \mathbf{K}_{S_i})\mathbf{v}_i)|| \\
&= ||\mathbf{K}^{\frac{1}{2}}(\mathbf{K}^\dagger \mathbf{y} - (\mathbf{K}_{S_i})^\dagger \mathbf{y} + y_i(\mathbf{K}_{S_i})^\dagger \mathbf{b} \\
&\quad + (\mathbf{I} - \mathbf{K}^\dagger \mathbf{K})(\mathbf{v} - \mathbf{v}_i) + (\mathbf{K}^\dagger \mathbf{K} - (\mathbf{K}_{S_i})^\dagger \mathbf{K}_{S_i})\mathbf{v}_i)|| \\
&= ||\mathbf{K}^{\frac{1}{2}}[(\mathbf{K}^\dagger - (\mathbf{K}_{S_i})^\dagger)\mathbf{y} + (\mathbf{I} - \mathbf{K}^\dagger \mathbf{K})(\mathbf{v} - \mathbf{v}_i) - (\mathbf{K}^\dagger \mathbf{K} - (\mathbf{K}_{S_i})^\dagger \mathbf{K}_{S_i})\mathbf{v}_i]||
\end{aligned}
\tag{10}
$$

Here we make use of the fact that $(\mathbf{K}_{S_i})^\dagger \mathbf{b} = \mathbf{0}$. If $\mathbf{K}$ has full rank (as in Remark 2), we see that $\mathbf{b}$ lies in the column space of $\mathbf{K}$ and $\mathbf{a}_*$ lies in the column space of $\mathbf{K}^\top$. Furthermore, $\beta_* = 1 + \mathbf{a}_*^\top \mathbf{K}^\dagger \mathbf{b} = 1 + \mathbf{a}^\top \mathbf{K}^\dagger \mathbf{b} + K(\mathbf{x}_i, \mathbf{x}_i)\mathbf{b}^\top \mathbf{K}^\dagger \mathbf{b} = \mathbf{K}_{ii}(\mathbf{K}^\dagger)_{ii} \neq 0$. Using equation 2.2 of Baksalary

et al. (2003) we obtain:

$$
\begin{aligned}
\mathbf{K}_*^\dagger &= \mathbf{K}^\dagger - (\mathbf{K}_{ii}(\mathbf{K}^\dagger)_{ii})^{-1}\mathbf{K}^\dagger \mathbf{b}\mathbf{a}_*^\top \mathbf{K}^\dagger \\
&= \mathbf{K}^\dagger - (\mathbf{K}_{ii}(\mathbf{K}^\dagger)_{ii})^{-1}\mathbf{K}^\dagger \mathbf{b}\mathbf{a}^\top \mathbf{K}^\dagger - ((\mathbf{K}^\dagger)_{ii})^{-1}\mathbf{K}^\dagger \mathbf{b}\mathbf{b}^\top \mathbf{K}^\dagger \\
&= \mathbf{K}^\dagger + (\mathbf{K}_{ii}(\mathbf{K}^\dagger)_{ii})^{-1}\mathbf{K}^\dagger \mathbf{b}\mathbf{b}^\top - ((\mathbf{K}^\dagger)_{ii})^{-1}\mathbf{K}^\dagger \mathbf{b}\mathbf{b}^\top \mathbf{K}^\dagger
\end{aligned}
\tag{11}
$$

Here we make use of the fact that $\mathbf{a}^\top \mathbf{K}^\dagger = -\mathbf{b}$. Also, using the corresponding formula from List 2 of Baksalary et al. (2003), we have $\mathbf{K}_*^\dagger \mathbf{K}_* = \mathbf{K}^\dagger \mathbf{K}$.

Next, we see that since $\mathbf{K}_*$ has the same rank as $\mathbf{K}$, $\mathbf{a}$ lies in the column space of $\mathbf{K}_*$, and $\mathbf{b}$ lies in the column space of $\mathbf{K}_*^\top$. Furthermore $\beta = 1 + \mathbf{b}^\top \mathbf{K}_* \mathbf{a} = 0$. This means we can use Theorem 6 in Meyer (1973) (equivalent to formula 2.1 in Baksalary et al. (2003)) to obtain the expression for $(\mathbf{K}_{S_i})^\dagger$, with $\mathbf{k} = \mathbf{K}_*^\dagger \mathbf{a}$ and $\mathbf{h} = \mathbf{b}^\top \mathbf{K}_*^\dagger$.

$$
\begin{aligned}
(\mathbf{K}_{S_i})^\dagger &= \mathbf{K}_*^\dagger - \mathbf{k}\mathbf{k}^\dagger \mathbf{K}_*^\dagger - \mathbf{K}_*^\dagger \mathbf{h}^\dagger \mathbf{h} + (\mathbf{k}^\dagger \mathbf{K}_*^\dagger \mathbf{h}^\dagger)\mathbf{k}\mathbf{h} \\
\implies (\mathbf{K}_{S_i})^\dagger - \mathbf{K}_*^\dagger &= (\mathbf{k}^\dagger \mathbf{K}_*^\dagger \mathbf{h}^\dagger)\mathbf{k}\mathbf{h} - \mathbf{k}\mathbf{k}^\dagger \mathbf{K}_*^\dagger - \mathbf{K}_*^\dagger \mathbf{h}^\dagger \mathbf{h} \\
\implies \|(\mathbf{K}_{S_i})^\dagger - \mathbf{K}_*^\dagger\|_{op} &\le 3\|\mathbf{K}_*^\dagger\|_{op}
\end{aligned}
\tag{12}
$$

Above, we use the fact that the operator norm of a rank 1 matrix is given by $\|\mathbf{u}\mathbf{v}^\top\|_{op} = \|\mathbf{u}\| \times \|\mathbf{v}\|$. Also, using the corresponding formula from List 2 of Baksalary et al. (2003), we have:

$$
\begin{aligned}
(\mathbf{K}_{S_i})^\dagger \mathbf{K}_{S_i} &= \mathbf{K}_*^\dagger \mathbf{K}_* - \mathbf{k}\mathbf{k}^\dagger \\
\implies \mathbf{K}^\dagger \mathbf{K} - (\mathbf{K}_{S_i})^\dagger \mathbf{K}_{S_i} &= \mathbf{k}\mathbf{k}^\dagger
\end{aligned}
\tag{13}
$$

Putting the two parts together we obtain the bound on $\left\|(\mathbf{K}_{S_i})^\dagger - \mathbf{K}^\dagger\right\|_{op}$:

$$
\begin{aligned}
\|\mathbf{K}^\dagger - (\mathbf{K}_{S_i})^\dagger\|_{op} &= \|\mathbf{K}^\dagger - \mathbf{K}_*^\dagger + \mathbf{K}_*^\dagger - (\mathbf{K}_{S_i})^\dagger\|_{op} \\
&\le 3\|\mathbf{K}_*^\dagger\|_{op} + \|\mathbf{K}^\dagger - \mathbf{K}_*^\dagger\|_{op} \\
&\le 3\|\mathbf{K}^\dagger\|_{op} + 4(\mathbf{K}_{ii}(\mathbf{K}^\dagger)_{ii})^{-1}\|\mathbf{K}^\dagger\|_{op} + 4((\mathbf{K}^\dagger)_{ii})^{-1}\|\mathbf{K}^\dagger\|_{op}^2 \\
&\le \|\mathbf{K}^\dagger\|_{op}(3 + 8\|\mathbf{K}^\dagger\|_{op}\|\mathbf{K}\|_{op})
\end{aligned}
\tag{14}
$$

The last step follows from $(\mathbf{K}_{ii})^{-1} \le \|\mathbf{K}^\dagger\|_{op}$ and $((\mathbf{K}^\dagger)_{ii})^{-1} \le \|\mathbf{K}\|_{op}$.

Plugging in these calculations into equation 10 we get:

$$
\begin{aligned}
\|\hat{f}_S - \hat{f}_{S_i}\|_{\mathcal{H}} &= \|\mathbf{K}^{\frac{1}{2}}[(\mathbf{K}^\dagger - (\mathbf{K}_{S_i})^\dagger)\mathbf{y} + (\mathbf{I} - \mathbf{K}^\dagger \mathbf{K})(\mathbf{v} - \mathbf{v}_i) - (\mathbf{K}^\dagger \mathbf{K} - (\mathbf{K}_{S_i})^\dagger \mathbf{K}_{S_i})\mathbf{v}_i]\| \\
&\le \|\mathbf{K}^{\frac{1}{2}}\|_{op}\left(\|(\mathbf{K}^\dagger - (\mathbf{K}_{S_i})^\dagger)\mathbf{y}\| + \|(\mathbf{I} - \mathbf{K}^\dagger \mathbf{K})(\mathbf{v} - \mathbf{v}_i)\| + \|\mathbf{k}\mathbf{k}^\dagger \mathbf{v}_i\|\right) \\
&\le \|\mathbf{K}^{\frac{1}{2}}\|_{op}(B_0 + \|\mathbf{I} - \mathbf{K}^\dagger \mathbf{K}\|_{op}\|\mathbf{v} - \mathbf{v}_i\| + \|\mathbf{v}_i\|)
\end{aligned}
\tag{15}
$$

We see that the right hand side is minimized when $\mathbf{v} = \mathbf{v}_i = \mathbf{0}$. We have also computed $B_0 = C \times \|\mathbf{K}^\dagger\|_{op} \times cond(\mathbf{K}) \times \|\mathbf{y}\|$, which concludes the proof of Lemma 11.

## 5 REMARK AND RELATED WORK

In the previous section we obtained bounds on the $CV_{loo}$ stability of interpolating solutions to the kernel least squares problem. Our kernel least squares results can be compared with stability bounds for regularized ERM (see Remark 3). Regularized ERM has a strong stability guarantee in terms of a uniform stability bound which turns out to be inversely proportional to the regularization parameter $\lambda$ and the sample size $n$ (Bousquet & Elisseeff, 2001). However, this estimate becomes vacuous as $\lambda \to 0$. In this paper, we establish a bound on average stability, and show that this bound is minimized when the minimum norm ERM solution is chosen. We study average stability since one can expect

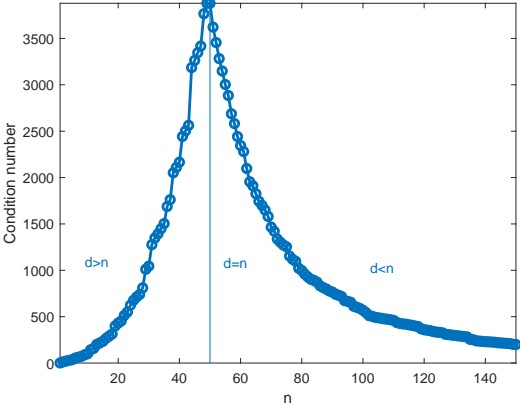

Figure 2: Typical double descent of the condition number (y axis) of a radial basis function kernel $K(x, x') = \exp\left(-\frac{||x-x'||^2}{2\sigma^2}\right)$ built from a random data matrix distributed as $\mathcal{N}(0, 1)$: as in the linear case, the condition number is worse when $n = d$, better if $n > d$ (on the right of $n = d$) and also better if $n < d$ (on the left of $n = d$). The parameter $\sigma$ was chosen to be 5. From Poggio et al. (2019)

worst case scenarios where the minimum norm is arbitrarily large (when $n \approx d$). One of our key findings is the relationship between minimizing the norm of the ERM solution and minimizing a bound on stability.

This leads to a second observation, namely, that we can consider the limit of our risk bounds as both the sample size ($n$) and the dimensionality of the data ($d$) go to infinity, but the ratio $\frac{d}{n} \to \gamma > 1$ as $n, d \to \infty$. This is a classical setting in statistics which allows us to use results from random matrix theory (Marchenko & Pastur, 1967). In particular, for linear kernels the behavior of the smallest eigenvalue of the kernel matrix (which appears in our bounds) can be characterized in this asymptotic limit. In fact, under appropriate distributional assumptions, our bound for linear regression can be computed as $(||\mathbf{X}^\dagger|| \times ||\mathbf{y}||)^2 \approx \frac{\sqrt{n}}{\sqrt{d}-\sqrt{n}} \to \frac{1}{\sqrt{\gamma}-1}$. Here the dimension of the data coincides with the number of parameters in the model. Interestingly, analogous results hold for more general kernels (inner product and RBF kernels) (El Karoui, 2010) where the asymptotics are taken with respect to the number and dimensionality of the data. These results predict a double descent curve for the condition number as found in practice, see Figure 2. While it may seem that our bounds in Theorem 7 diverge if $d$ is held constant and $n \to \infty$, this case is not covered by our theorem, since when $n > d$ we no longer have interpolating solutions.

Recently, there has been a surge of interest in studying linear and kernel least squares models, since classical results focus on situations where constraints or penalties that prevent interpolation are added to the empirical risk. For example, high dimensional linear regression is considered in Mei & Montanari (2019); Hastie et al. (2019); Bartlett et al. (2019), and "ridgeless" kernel least squares is studied in Liang et al. (2019); Rakhlin & Zhai (2018) and Liang et al. (2020). While these papers study upper and lower bounds on the risk of interpolating solutions to the linear and kernel least squares problem, ours are the first to derived using stability arguments. While it might be possible to obtain tighter excess risk bounds through careful analysis of the minimum norm interpolant, our simple approach helps us establish a link between stability in statistical and in numerical sense.

Finally, we can compare our results with observations made in Poggio et al. (2019) on the condition number of random kernel matrices. The condition number of the empirical kernel matrix is known to control the numerical stability of the solution to a kernel least squares problem. Our results show that the statistical stability is also controlled by the condition number of the kernel matrix, providing a natural link between numerical and statistical stability.

## 6    CONCLUSIONS

In summary, minimizing a bound on cross validation stability minimizes the expected error in both the classical and the modern regime of ERM. In the classical regime ($d < n$), $CV_{loo}$ stability implies generalization and consistency for $n \to \infty$. In the modern regime ($d > n$), as described in this paper, $CV_{loo}$ stability can account for the double descent curve in kernel interpolants (Belkin et al., 2019) under appropriate distributional assumptions. The main contribution of this paper is characterizing stability of interpolating solutions, in particular deriving excess risk bounds via a stability argument. In the process, we show that among all the interpolating solutions, the one with minimum norm also minimizes a bound on stability. Since the excess risk bounds of the minimum norm interpolant depend on the pseudoinverse of the kernel matrix, we establish here an elegant link between *numerical and statistical* stability. This also holds for solutions computed by gradient descent, since gradient descent converges to minimum norm solutions in the case of "linear" kernel methods. Our approach is simple and combines basic stability results with matrix inequalities.

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

## A  EXCESS RISK, GENERALIZATION, AND STABILITY

We use the same notation as introduced in Section 2 for the various quantities considered in this section. That is in the supervised learning setup $V(f, z)$ is the loss incurred by hypothesis $f$ on the sample $z$, and $I[f] = \mathbb{E}_z[V(f, z)]$ is the expected error of hypothesis $f$. Since we are interested in different forms of stability, we will consider learning problems over the original training set $S = \{z_1, z_2, \ldots, z_n\}$, the leave one out training set $S_i = \{z_1, \ldots, z_{i-1}, z_{i+1}, \ldots, z_n\}$, and the replace one training set $(S_i, z) = \{z_1, \ldots, z_{i-1}, z_{i+1}, \ldots, z_n, z\}$

### A.1  REPLACE ONE AND LEAVE ONE OUT ALGORITHMIC STABILITY

Similar to the definition of expected $\text{CV}_{loo}$ stability in equation (3) of the main paper, we say an algorithm is cross validation *replace one* stable (in expectation), denoted as $\text{CV}_{ro}$, if there exists $\beta_{ro} > 0$ such that

$$\mathbb{E}_{S,z}[V(f_S, z) - V(f_{(S_i,z)}, z)] \leq \beta_{ro}.$$

We can strengthen the above stability definition by introducing the notion of replace one algorithmic stability (in expectation) Bousquet & Elisseeff (2001). There exists $\alpha_{ro} >$ such that for all $i = 1, \ldots, n$,

$$\mathbb{E}_{S,z}[\|f_S - f_{(S_i,z)}\|_\infty] \leq \alpha_{ro}.$$

We make two observations:
First, if the loss is Lipschitz, that is if there exists $C_V > 0$ such that for all $f, f' \in \mathcal{H}$

$$\|V(f, z) - V(f', z)\| \leq C_V \|f - f'\|,$$

then replace one algorithmic stability implies $\text{CV}_{ro}$ stability with $\beta_{ro} = C_V \alpha_{ro}$. Moreover, the same result holds if the loss is locally Lipschitz and there exists $R > 0$, such that $\|f_S\|_\infty \leq R$ almost surely. In this latter case the Lipschitz constant will depend on $R$. Later, we illustrate this situation for the square loss.

Second, we have for all $i = 1, \ldots, n$, $S$ and $z$,

$$\mathbb{E}_{S,z}[\|f_S - f_{(S_i,z)}\|_\infty] \leq \mathbb{E}_{S,z}[\|f_S - f_{S_i}\|_\infty] + \mathbb{E}_{S,z}[\|f_{(S_i,z)} - f_{S_i}\|_\infty].$$

This observation motivates the notion of leave one out algorithmic stability (in expectation) Bousquet & Elisseeff (2001)]

$$\mathbb{E}_{S,z}[\|f_S - f_{S_i}\|_\infty] \leq \alpha_{loo}.$$

Clearly, leave one out algorithmic stability implies replace one algorithmic stability with $\alpha_{ro} = 2\alpha_{loo}$ and it implies also $\text{CV}_{ro}$ stability with $\beta_{ro} = 2C_V\alpha_{loo}$.

### A.2  EXCESS RISK AND $\text{CV}_{loo}$, $\text{CV}_{ro}$ STABILITY

We recall the statement of Lemma 5 in section 3 that bounds the excess risk using the $\text{CV}_{loo}$ stability of a solution.

**Lemma 13 (Excess Risk & $\text{CV}_{loo}$ Stability)**  *For all $i = 1, \ldots, n$,*

$$\mathbb{E}_S[I[f_{S_i}] - \inf_{f \in \mathcal{H}} I[f]] \leq \mathbb{E}_S[V(f_{S_i}, z_i) - V(f_S, z_i)]. \tag{16}$$

In this section, two properties of ERM are useful, namely symmetry, and a form of unbiasedness.

**Symmetry.**  A key property of ERM is that it is *symmetric* with respect to the data set $S$, meaning that it does not depend on the order of the data in $S$.

A second property relates the expected ERM with the minimum of expected risk.

**ERM Bias.** The following inequality holds.

$$\mathbb{E}[[I_S[f_S]] - \min_{f \in \mathcal{H}} I[f] \leq 0. \tag{17}$$

To see this, note that

$$I_S[f_S] \leq I_S[f]$$

for all $f \in \mathcal{H}$ by definition of ERM, so that taking the expectation of both sides

$$\mathbb{E}_S[I_S[f_S]] \leq \mathbb{E}_S[I_S[f]] = I[f]$$

for all $f \in \mathcal{H}$. This implies

$$\mathbb{E}_S[I_S[f_S]] \leq \min_{f \in \mathcal{H}} I[f]$$

and hence (17) holds.

**Remark 14** *Note that the same argument gives more generally that*

$$\mathbb{E}[\inf_{f \in \mathcal{H}}[I_S[f]] - \inf_{f \in \mathcal{H}} I[f] \leq 0. \tag{18}$$

Given the above premise, the proof of Lemma 5 is simple.

**Proof** [of Lemma 5] Adding and subtracting $\mathbb{E}_S[I_S[f_S]]$ from the expected excess risk we have that

$$\mathbb{E}_S[I[f_{S_i}] - \min_{f \in \mathcal{H}} I[f]] = \mathbb{E}_S[I[f_{S_i}] - I_S[f_S] + I_S[f_S] - \min_{f \in \mathcal{H}} I[f]], \tag{19}$$

and since $\mathbb{E}_S[I_S[f_S]] - \min_{f \in \mathcal{H}} I[f]$ is less or equal than zero, see (18), then

$$\mathbb{E}_S[I[f_{S_i}] - \min_{f \in \mathcal{H}} I[f]] \leq \mathbb{E}_S[I[f_{S_i}] - I_S[f_S]]. \tag{20}$$

Moreover, for all $i = 1, \ldots, n$

$$\mathbb{E}_S[I[f_{S_i}]] = \mathbb{E}_S[\mathbb{E}_{z_i} V(f_{S_i}, z_i)] = \mathbb{E}_S[V(f_{S_i}, z_i)]$$

and

$$\mathbb{E}_S[I_S[f_S]] = \frac{1}{n} \sum_{i=1}^{n} \mathbb{E}_S[V(f_S, z_i)] = \mathbb{E}_S[V(f_S, z_i)].$$

Plugging these last two expressions in (20) and in (19) leads to (4). $\blacksquare$

We can prove a similar result relating excess risk with $\text{CV}_{ro}$ stability.

**Lemma 15 (Excess Risk & $\text{CV}_{ro}$ Stability)** *Given the above definitions, the following inequality holds for all $i = 1, \ldots, n$,*

$$\mathbb{E}_S[I[f_S] - \inf_{f \in \mathcal{H}} I[f]] \leq \mathbb{E}_S[I[f_S] - I_S[f_S]] = \mathbb{E}_{S,z}[V(f_S, z) - V(f_{(S_i, z)}, z)]. \tag{21}$$

**Proof** The first inequality is clear from adding and subtracting $I_S[f_S]$ from the expected risk $I[f_S]$ we have that

$$\mathbb{E}_S[I[f_S] - \min_{f \in \mathcal{H}} I[f]] = \mathbb{E}_S[I[f_S] - I_S[f_S] + I_S[f_S] - \min_{f \in \mathcal{H}} I[f]],$$

and recalling (18). The main step in the proof is showing that for all $i = 1, \ldots, n$,

$$\mathbb{E}[I_S[f_S]] = \mathbb{E}[V(f_{(S_i, z)}, z)] \tag{22}$$

to be compared with the trivial equality, $\mathbb{E}[I_S[f_S] = \mathbb{E}[V(f_S, z_i)]$. To prove Equation (22), we have for all $i = 1, \ldots, n$,

$$\mathbb{E}_S[I_S[f_S]] = \mathbb{E}_{S,z}[\frac{1}{n} \sum_{i=1}^{n} V(f_S, z_i)] = \frac{1}{n} \sum_{i=1}^{n} \mathbb{E}_{S,z}[V(f_{(S_i, z)}, z)] = \mathbb{E}_{S,z}[V(f_{(S_i, z)}, z)]$$

where we used the fact that by the symmetry of the algorithm $\mathbb{E}_{S,z}[V(f_{(S_i, z)}, z)]$ is the same for all $i = 1, \ldots, n$. The proof is concluded noting that $\mathbb{E}_S[I[f_S]] = \mathbb{E}_{S,z}[V(f_S, z)]$. $\blacksquare$

A.3 DISCUSSION ON STABILITY AND GENERALIZATION

Below we discuss some more aspects of stability and its connection to other quantities in statistical learning theory.

**Remark 16 (CV$_{loo}$ stability in expectation and in probability)** *In Mukherjee et al. (2006), CV$_{loo}$ stability is defined in probability, that is there exists $\beta_{CV}^P > 0$, $0 < \delta_{CV}^P \leq 1$ such that*

$$\mathbb{P}_S\{|V(f_{S_i}, z_i) - V(f_S, z_i)| \geq \beta_{CV}^P\} \leq \delta_{CV}^P.$$

*Note that the absolute value is not needed for ERM since almost positivity holds Mukherjee et al. (2006), that is $V(f_{S_i}, z_i) - V(f_S, z_i) > 0$. Then CV$_{loo}$ stability in probability and in expectation are clearly related and indeed equivalent for bounded loss functions. CV$_{loo}$ stability in expectation (3) is what we study in the following sections.*

**Remark 17 (Connection to uniform stability and other notions of stability)** *Uniform stability, introduced by Bousquet & Elisseeff (2001), corresponds in our notation to the assumption that there exists $\beta_u > 0$ such that for all $i = 1, \ldots, n$, $\sup_{z \in Z} |V(f_{S_i}, z) - V(f_S, z)| \leq \beta_u$. Clearly this is a strong notion implying most other definitions of stability. We note that there are number of different notions of stability. We refer the interested reader to Kutin & Niyogi (2002), Mukherjee et al. (2006).*

**Remark 18 (CV$_{loo}$ Stability & Learnability)** *A natural question is to which extent suitable notions of stability are not only sufficient but also necessary for controlling the excess risk of ERM. Classically, the latter is characterized in terms of a uniform version of the law of large numbers, which itself can be characterized in terms of suitable complexity measures of the hypothesis class. Uniform stability is too strong to characterize consistency while CV$_{loo}$ stability turns out to provide a suitably weak definition as shown in Mukherjee et al. (2006), see also Kutin & Niyogi (2002), Mukherjee et al. (2006). Indeed, a main result in Mukherjee et al. (2006) shows that CV$_{loo}$ stability is equivalent to consistency of ERM:*

**Theorem 19** *Mukherjee et al. (2006) For ERM and bounded loss functions, CV$_{loo}$ stability in probability with $\beta_{CV}^P$ converging to zero for $n \to \infty$ is equivalent to consistency and generalization of ERM.*

**Remark 20 (CV$_{loo}$ stability & in-sample/out-of-sample error)** *Let $(S, z) = \{z_1, \ldots, z_n, z\}$, ($z$ is a data point drawn according to the same distribution) and the corresponding ERM solution $f_{(S,z)}$, then (4) can be equivalently written as,*

$$\mathbb{E}_S[I[f_S] - \inf_{f \in \mathcal{F}} I[f]] \leq \mathbb{E}_{S,z}[V(f_S, z) - V(f_{(S,z)}, z)].$$

*Thus CV$_{loo}$ stability measures how much the loss changes when we test on a point that is present in the training set and absent from it. In this view, it can be seen as an average measure of the difference between in-sample and out-of-sample error.*

**Remark 21 (CV$_{loo}$ stability and generalization)** *A common error measure is the (expected) generalization gap $\mathbb{E}_S[I[f_S] - I_S[f_S]]$. For non-ERM algorithms, CV$_{loo}$ stability by itself not sufficient to control this term, and further conditions are needed Mukherjee et al. (2006), since*

$$\mathbb{E}_S[I[f_S] - I_S[f_S]] = \mathbb{E}_S[I[f_S] - I_S[f_{S_i}]] + \mathbb{E}_S[I_S[f_{S_i}] - I_S[f_S]].$$

*The second term becomes for all $i = 1, \ldots, n$,*

$$\mathbb{E}_S[I_S[f_{S_i}] - I_S[f_S]] = \frac{1}{n} \sum_{i=1}^{n} \mathbb{E}_S[V(f_{S_i}, z_i) - V(f_S, z_i)] = \mathbb{E}_S[V(f_{S_i}, z_i) - V(f_S, z_i)]$$

*and hence is controlled by CV stability. The first term is called expected leave one out error in Mukherjee et al. (2006) and is controlled in ERM as $n \to \infty$, see Theorem 19 above.*

## B CV$_{loo}$ STABILITY OF LINEAR REGRESSION

We have a dataset $S = \{(\mathbf{x}_i, y_i)\}_{i=1}^n$ and we want to find a mapping $\mathbf{w} \in \mathbb{R}^d$, that minimizes the empirical least squares risk. All interpolating solutions are of the form $\hat{\mathbf{w}}_S = \mathbf{y}^\top \mathbf{X}^\dagger + \mathbf{v}^\top (\mathbf{I} - \mathbf{X}\mathbf{X}^\dagger)$. Similarly, all interpolating solutions on the leave one out dataset $S_i$ can be written as $\hat{\mathbf{w}}_{S_i} = \mathbf{y}_i^\top (\mathbf{X}_i)^\dagger + \mathbf{v}_i^\top (\mathbf{I} - \mathbf{X}_i(\mathbf{X}_i)^\dagger)$. Here $\mathbf{X}, \mathbf{X}_i \in \mathbb{R}^{d \times n}$ are the data matrices for the original and leave one out datasets respectively. We note that when $\mathbf{v} = \mathbf{0}$ and $\mathbf{v}_i = \mathbf{0}$, we obtain the minimum norm interpolating solutions on the datasets $S$ and $S_i$.

In this section we want to estimate the CV$_{loo}$ stability of the minimum norm solution to the ERM problem in the linear regression case. This is the case outlined in Remark 9 of the main paper. In order to prove Remark 9, we only need to combine Lemma 10 with the linear regression analogue of Lemma 11. We state and prove that result in this section. This result predicts a double descent curve for the norm of the pseudoinverse as found in practice, see Figure 3.

**Lemma 22** *Let $\hat{\mathbf{w}}_S, \hat{\mathbf{w}}_{S_i}$ be any interpolating least squares solutions on the full and leave one out datasets $S, S_i$, then $\|\hat{\mathbf{w}}_S - \hat{\mathbf{w}}_{S_i}\| \leq B_{CV}(\mathbf{X}^\dagger, \mathbf{y}, \mathbf{v}, \mathbf{v}_i)$, and $B_{CV}$ is minimized when $\mathbf{v} = \mathbf{v}_i = \mathbf{0}$, which corresponds to the minimum norm interpolating solutions $\mathbf{w}_S^\dagger, \mathbf{w}_{S_i}^\dagger$.*

*Also,*

$$\left\| \mathbf{w}_S^\dagger - \mathbf{w}_{S_i}^\dagger \right\| \leq 3 \left\| \mathbf{X}^\dagger \right\|_{op} \times \|\mathbf{y}\| \tag{23}$$

As mentioned before in section 2.1 of the main paper, linear regression can be viewed as a case of the kernel regression problem where $\mathcal{H} = \mathbb{R}^d$, and the feature map $\Phi$ is the identity map. The inner product and norms considered in this case are also the usual Euclidean inner product and 2-norm for vectors in $\mathbb{R}^d$. The notation $\|\cdot\|$ denotes the Euclidean norm for vectors both in $\mathbb{R}^d$ and $\mathbb{R}^n$. The usage of the norm should be clear from the context. Also, $\|\mathbf{A}\|_{op}$ is the left operator norm for a matrix $\mathbf{A} \in \mathbb{R}^{n \times d}$, that is $\|\mathbf{A}\|_{op} = \sup_{\mathbf{y} \in \mathbb{R}^n, \|\mathbf{y}\| = 1} \|\mathbf{y}^\top \mathbf{A}\|$.

We have $n$ samples in the training set for a linear regression problem, $\{(\mathbf{x}_i, y_i)\}_{i=1}^n$. We collect all the samples into a single matrix/vector $\mathbf{X} = [\mathbf{x}_1 \mathbf{x}_2 \ldots \mathbf{x}_n] \in \mathbb{R}^{d \times n}$, and $\mathbf{y} = [y_1 y_2 \ldots y_n]^\top \in \mathbb{R}^n$. Then any interpolating ERM solution $\mathbf{w}_S$ satisfies the linear equation

$$\mathbf{w}_S^\top \mathbf{X} = \mathbf{y}^\top \tag{24}$$

Any interpolating solution can be written as:

$$(\hat{\mathbf{w}}_S)^\top = \mathbf{y}^\top \mathbf{X}^\dagger + \mathbf{v}^\top (\mathbf{I} - \mathbf{X}\mathbf{X}^\dagger). \tag{25}$$

If we consider the leave one out training set $S_i$ we can find the minimum norm ERM solution for $\mathbf{X}_i = [\mathbf{x}_1 \ldots \mathbf{0} \ldots \mathbf{x}_n]$ and $\mathbf{y}_i = [y_1 \ldots 0 \ldots y_n]^\top$ as

$$(\hat{\mathbf{w}}_{S_i})^\top = \mathbf{y}_i^\top (\mathbf{X}_i)^\dagger + \mathbf{v}_i^\top (\mathbf{I} - \mathbf{X}_i(\mathbf{X}_i)^\dagger). \tag{26}$$

We can write $\mathbf{X}_i$ as:

$$\mathbf{X}_i = \mathbf{X} + \mathbf{a}\mathbf{b}^\top \tag{27}$$

where $\mathbf{a} \in \mathbb{R}^d$ is a column vector representing the additive change to the $i^{\text{th}}$ column, i.e, $\mathbf{a} = -\mathbf{x}_i$, and $\mathbf{b} \in \mathbb{R}^{n \times 1}$ is the $i-$th element of the canonical basis in $\mathbb{R}^n$ (all the coefficients are zero but the $i-$th which is one). Thus $\mathbf{a}\mathbf{b}^\top$ is a $d \times n$ matrix composed of all zeros apart from the $i^{\text{th}}$ column which is equal to $\mathbf{a}$.

We also have $\mathbf{y}_i = \mathbf{y} - y_i \mathbf{b}$. Now per Lemma 10 we are interested in bounding the quantity $\|\hat{\mathbf{w}}_{S_i} - \hat{\mathbf{w}}_S\| = \|(\hat{\mathbf{w}}_{S_i})^\top - (\hat{\mathbf{w}}_S)^\top\|$. This simplifies to:

$$
\begin{aligned}
\|\hat{\mathbf{w}}_{S_i} - \hat{\mathbf{w}}_S\| &= \|\mathbf{y}_i^\top (\mathbf{X}_i)^\dagger - \mathbf{y}^\top \mathbf{X}^\dagger + \mathbf{v}_i^\top - \mathbf{v}^\top + \mathbf{v}^\top \mathbf{X}\mathbf{X}^\dagger - \mathbf{v}_i^\top \mathbf{X}_i(\mathbf{X}_i)^\dagger\| \\
&= \|(\mathbf{y}^\top - y_i \mathbf{b}^\top)(\mathbf{X}_i)^\dagger - \mathbf{y}^\top \mathbf{X}^\dagger + \mathbf{v}_i^\top - \mathbf{v}^\top + \mathbf{v}^\top \mathbf{X}\mathbf{X}^\dagger - \mathbf{v}_i^\top \mathbf{X}_i(\mathbf{X}_i)^\dagger\| \\
&= \|\mathbf{y}^\top ((\mathbf{X}_i)^\dagger - \mathbf{X}^\dagger) + y_i \mathbf{b}^\top (\mathbf{X}_i)^\dagger + \mathbf{v}_i^\top - \mathbf{v}^\top + \mathbf{v}^\top \mathbf{X}\mathbf{X}^\dagger - \mathbf{v}_i^\top \mathbf{X}_i(\mathbf{X}_i)^\dagger\| \quad (28) \\
&= \|\mathbf{y}^\top ((\mathbf{X}_i)^\dagger - \mathbf{X}^\dagger) + \mathbf{v}_i^\top - \mathbf{v}^\top + \mathbf{v}^\top \mathbf{X}\mathbf{X}^\dagger - \mathbf{v}_i^\top \mathbf{X}_i(\mathbf{X}_i)^\dagger\| \\
&= \|\mathbf{y}^\top ((\mathbf{X}_i)^\dagger - \mathbf{X}^\dagger) + (\mathbf{v}_i^\top - \mathbf{v}^\top)(\mathbf{I} - \mathbf{X}\mathbf{X}^\dagger) - \mathbf{v}_i^\top (\mathbf{X}\mathbf{X}^\dagger - \mathbf{X}_i(\mathbf{X}_i)^\dagger)\|
\end{aligned}
$$

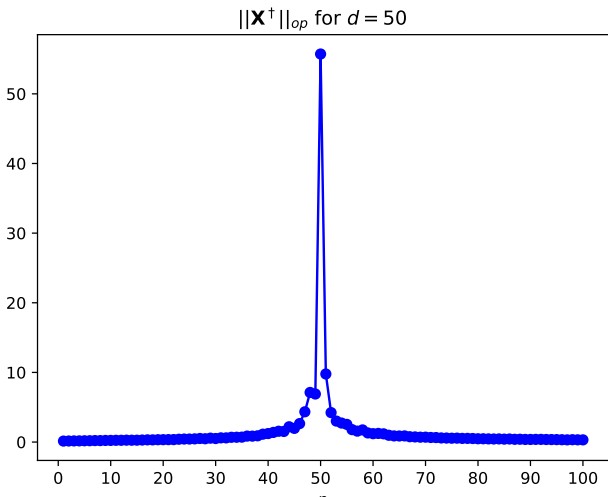

Figure 3: Typical double descent of the pseudoinverse norm (y axis) of a random data matrix distributed as $\mathcal{N}(0, 1)$: the condition number is worse when $n = d$, better if $n > d$ (on the right of $n = d$) and also better if $n < d$ (on the left of $n = d$).. From Poggio et al. (2019)

In the above equation we make use of the fact that $\mathbf{b}^\top (\mathbf{X}_i)^\dagger = \mathbf{0}$. We use an old formula (Meyer, 1973; Baksalary et al., 2003) to compute $(\mathbf{X}_i)^\dagger$ from $\mathbf{X}^\dagger$. We use the development of pseudo-inverses of perturbed matrices in Meyer (1973). We see that $\mathbf{a} = -\mathbf{x}_i$ is a vector in the column space of $\mathbf{X}$ and $\mathbf{b}$ is in the range space of $\mathbf{X}^T$ (provided $\mathbf{X}$ has full column rank), with $\beta = 1 + \mathbf{b}^\top \mathbf{X}^\dagger \mathbf{a} = 1 - \mathbf{b}^\top \mathbf{X}^\dagger \mathbf{x}_i = 0$. This means we can use Theorem 6 in Meyer (1973) (equivalent to formula 2.1 in Baksalary et al. (2003)) to obtain the expression for $(\mathbf{X}_i)^\dagger$

$$(\mathbf{X}_i)^\dagger = \mathbf{X}^\dagger - \mathbf{k}\mathbf{k}^\dagger \mathbf{X}^\dagger - \mathbf{X}^\dagger \mathbf{h}^\dagger \mathbf{h} + (\mathbf{k}^\dagger \mathbf{X}^\dagger \mathbf{h}^\dagger)\mathbf{k}\mathbf{h} \tag{29}$$

where $\mathbf{k} = \mathbf{X}^\dagger \mathbf{a}$, and $\mathbf{h} = \mathbf{b}^\top \mathbf{X}^\dagger$, and $\mathbf{u}^\dagger = \frac{\mathbf{u}^\top}{||\mathbf{u}||^2}$ for any non-zero vector $\mathbf{u}$.

$$\begin{aligned}
(\mathbf{X}_i)^\dagger - \mathbf{X}^\dagger &= (\mathbf{k}^\dagger \mathbf{X}^\dagger \mathbf{h}^\dagger)\mathbf{k}\mathbf{h} - \mathbf{k}\mathbf{k}^\dagger \mathbf{X}^\dagger - \mathbf{X}^\dagger \mathbf{h}^\dagger \mathbf{h} \\
&= \mathbf{a}^\top (\mathbf{X}^\dagger)^\top \mathbf{X}^\dagger (\mathbf{X}^\dagger)^\top \mathbf{b} \times \frac{\mathbf{k}\mathbf{h}}{||\mathbf{k}||^2 ||\mathbf{h}||^2} - \mathbf{k}\mathbf{k}^\dagger \mathbf{X}^\dagger - \mathbf{X}^\dagger \mathbf{h}^\dagger \mathbf{h} \\
\implies ||(\mathbf{X}_i)^\dagger - \mathbf{X}^\dagger||_{op} &\leq \frac{|\mathbf{a}^\top (\mathbf{X}^\dagger)^\top \mathbf{X}^\dagger (\mathbf{X}^\dagger)^\top \mathbf{b}|}{||\mathbf{X}^\dagger \mathbf{a}|| ||\mathbf{b}^\top \mathbf{X}^\dagger||} + 2||\mathbf{X}^\dagger||_{op} \\
&\leq \frac{||\mathbf{X}^\dagger||_{op} ||\mathbf{X}^\dagger \mathbf{a}|| ||\mathbf{b}^\top \mathbf{X}^\dagger||}{||\mathbf{X}^\dagger \mathbf{a}|| ||\mathbf{b}^\top \mathbf{X}^\dagger||} + 2||\mathbf{X}^\dagger||_{op} \\
&= 3||\mathbf{X}^\dagger||_{op}
\end{aligned} \tag{30}$$

The above set of inequalities follows from the fact that the operator norm of a rank 1 matrix is given by $||\mathbf{u}\mathbf{v}^\top||_{op} = ||\mathbf{u}|| \times ||\mathbf{v}||$

Also, from List 2 of Baksalary et al. (2003) we have that $\mathbf{X}_i(\mathbf{X}_i)^\dagger = \mathbf{X}\mathbf{X}^\dagger - \mathbf{h}^\dagger \mathbf{h}$.

Plugging in these calculations into equation 28 we get:

$$\begin{aligned}
||\hat{\mathbf{w}}_{S_i} - \hat{\mathbf{w}}_S|| &= ||\mathbf{y}^\top ((\mathbf{X}_i)^\dagger - \mathbf{X}^\dagger) + (\mathbf{v}_i^\top - \mathbf{v}^\top)(\mathbf{I} - \mathbf{X}\mathbf{X}^\dagger) - \mathbf{v}_i^\top (\mathbf{X}\mathbf{X}^\dagger - \mathbf{X}_i(\mathbf{X}_i)^\dagger)|| \\
&\leq B_0 + ||\mathbf{I} - \mathbf{X}\mathbf{X}^\dagger||_{op} ||\mathbf{v} - \mathbf{v}_i|| + ||\mathbf{v}_i|| \times ||\mathbf{h}^\dagger \mathbf{h}||_{op} \\
&\leq B_0 + 2||\mathbf{v} - \mathbf{v}_i|| + ||\mathbf{v}_i||
\end{aligned} \tag{31}$$

We see that the right hand side is minimized when $\mathbf{v} = \mathbf{v}_i = \mathbf{0}$. We can also compute $B_0 = 3||\mathbf{X}^\dagger||_{op} ||\mathbf{y}||$, which concludes the proof of Lemma 22.

