# OpenReview forum: "For interpolating kernel machines, minimizing the norm of the ERM solution minimizes stability"
_ICLR.cc/2021/Conference — Reject_

### Official Review · AnonReviewer4 · 2020-10-19
**For interpolating kernel machines, minimizing the norm of the ERM solution minimizes stability**

**Rating:** 6
**Confidence:** 3

**Review:**

In this paper, they provide the risk bounds of  kernel ridge-less regression (the regularization $\lambda-\rightarrow 0$) based on the CV_{loo} stability. They show  that the interpolating solution with minimum norm is the minimal bound of CV_{loo} stability, and can be controlled by the condition number of the empirical kernel matrix, which establishes an  elegant link between numerical and statistical stability.

Pros:
-1:  The stability of Tikhonov regularization has been well studied, but the study for the unregularized regression probelms is lacking. This paper fills the gap for unregularized regression.

-2:  They provide an upper bound on the stability of interpolating solutions, and show that among all interpolating solutions, the minimum norm solution has the best test error. They further provide emprical experiments to verify their theoretical findings.

-3: They show that CV_{loo} stability can be bounded by the condition number of the empirical kernel matrix, which establishes an  elegant link between numerical and statistical stability.

Cons:
In Theorem 7, the CV_{loo} stability of minimum norm solutions  can be bound by $C_1\beta_1+C\beta_2$.  Note that the kernel matrix $\mathbf K$ converges to the integral operator $L_K:  L_K(f)(x)=\int_{X}f(x)K(x,y)dy$ when $n\rightarrow \infty$, so $\|{\mathbf K}^{1/2}\|$  and $\|{\mathbf K}^+\|$  may converge to  two constants.  Note that the  condition number of the empirical kernel matrix $cond(\mathbf K)\geq 1$,  and $\|y\|$ grows with the increase of $n$, thus the $\beta_1$ and $\beta_2$ may grow  with the the increase of $n$. However, intuitively,  we think the CV_{loo} stability should decrease with the increase of $n$. So, we think the upper bound proposed in this paper may be too loose.

If you can answer my concern at this point, I will increase my score.

---

> ### Author Response · Authors · 2020-11-18
> **Response to AnonReviewer4**
>
> Thank you for your comments and constructive criticism of our work. It is correct that the condition number grows to infinity for fixed $d$ and $n \to \infty$. Thus we agree with you that if $d$ is held constant and $n$ is allowed to grow unbounded, our upper bound would also diverge and thus turn out to be vacuous. But, as we mentioned in the answer to reviewer 1, the solutions to the linear problem do not interpolate when $n>d$. As a matter of fact, our results (Theorem 7 in particular) are meant to hold in the regime where $\frac{d}{n} \rightarrow \gamma>1$ as $d,n \rightarrow \infty$. In this regime, the spectral norm and condition number of $\mathbf{K}$ and the norm of $\mathbf{K^\dagger y}$ do not diverge. While our results do not require any distributional assumptions, we can see how the bounds do not diverge under the assumption of random data. In the linear case, under appropriate distributional assumptions, we have $||\mathbf{X}^\dagger||\times ||\mathbf{y}|| \approx \frac{\sqrt{n}}{\sqrt{d} - \sqrt{n}} \rightarrow \frac{1}{\sqrt{\gamma}-1}$. For random data, the extension to the kernel case also holds since by the results of El Karoui [EK], an inner product/RBF kernel can be approximated by its linear term. We will include the above discussion in an updated draft of the paper.
>
> [EK] El Karoui, N. (2010). The spectrum of kernel random matrices. The Annals of Statistics, 38(1), 1-50.

---

### Official Review · AnonReviewer2 · 2020-10-20
**Paper makes a worthwhile contribution and is to the point. The authors might, however, have overlooked something crucial which leads to the paper having more complex results than necessary.**

**Rating:** 8
**Confidence:** 4

**Review:**

UPDATE:
As the authors were already aware of the zero loss case and analyzed this previously, I am confident that the authors can address this to the point in an updated version. With this I think this is a good paper that should be accepted.

##########################

Summary:
The papers addresses the setting of overparametrized models that interpolate the training data, and the related double descent observation in a kernel setting. The overparametrized case of interpolating models is not yet that well understood, but of importance as the success of neural networks is closely related to that setting. This paper shows that the minimum norm interpolating solution is optimal (among all interpolating solutions) with respect to a derived bound on the expected leave-one-out stability, and thus also optimal in the same sense with respect to the excess risk.

##########################

##########################

Pros:
The paper is well written, to the point, and technically (mostly! see cons) sound.
To the best of my knowledge this particular stability analysis is novel, and thus warrants a publication, in particular as the overparametrized case is not that well understood as of yet.

##########################

##########################

Cons:
I actually don't have many cons, I enjoyed the paper. There is however one thing that the authors seemed to have missed:
The term V(hat(f_S),z_i) is zero, as hat(f_S) interpolates the training data and z_i is part of it. That doesn't mean that any of the theory is wrong, but that creates two problems in my opinion:
1. The story about leave-one-out stability does not make sense anymore. In fact the expected leave-one-out stability is just the expected risk for interpolating solutions.
2. I would imagine that most of the results can be simplified because of that. I could imagine that all the results hold with essentially all terms regarding hat(f_s) being removed. I think the qualitative conclusions would remain the same though.

My suggestion would be to leave the paper as is (the results as far as I see also hold for not interpolating solutions), and then discuss the interpolating solutions as an extra case.

##########################

##########################

Scoring:
For now I will have to vote for a rejection, as I am not sure if the problem that I mention can be addressed in one rebuttal phase. But I am happily convinced otherwise, or convinced that I am wrong in any other way.

##########################

##########################

Additional feedback:
- When I first read the title I thought that you wanted to show minimal norm solutions are NOT stable, as it has 'minimal' stability. I understand now that minimal refers to the numerical value of the stability definition, but still the wording was somewhat confusing. (Just to consider, no need to change for me if you think it is correct like that)

- Equation (2) and also bit later. You use a comma to separate an index "S,i", fairly unusual I would say.

- Remark 4, rework first sentence.

- Equation (7), in the very basic property of RKHS the kappa would depend on x, for you that seems to follow with one of your assumptions + cauchy-schwarz. I would not consider that a basic property.

##########################

---

> ### Author Response · Authors · 2020-11-18
> **Response to AnonReviewer2**
>
> We thank you for your comments and constructive criticism.
>
> 1. Indeed the analysis holds with and without zero loss but can be simplified/sharpened in the former case. In an early version of the paper, we actually started with the analysis of  this "zero loss case" and then switched to the more general case because it entails essentially the same calculations.  The proof of our main theorem (Theorem 7) on bounding leave one-out stability results from combining Lemmas 10 and 11. In Lemma 10 we use the locally Lipschitz property of the squared loss function to bound the leave one out stability in terms of the difference between the norms of the solutions. In Lemma 11 we obtain a bound on the difference between the norms of the solutions $||\hat{f}_{S_i} - \hat{f}_S||_\mathcal{H}.$  If we set the term $V(\hat{f}_S \, z_i )=0$ (the "zero-loss" case), the leave one out stability reduces to $\mathbb{E}_S \left[ V(  \hat{f}_\{S_i\} \, z_i) - V( \hat{f}_S \, z_i) \right] = \mathbb{E}_S \left[ V(  \hat{f}_\{S_i\} \, z_i) \right] = \mathbb{E}_S \left[ (  \hat{f}_\{S_i\} (\mathbf{x}_i) - y_i )^2 \right] = \mathbb{E}_S \left[ (  \hat{f}_\{S_i\} (\mathbf{x}_i) - \hat{f}_S (\mathbf{x}_i))^2 \right].$ We can use the RKHS property to get the leave one out stability $= \mathbb{E}_S \left[ \langle  \hat{f}_\{S_i\} (\cdot) - \hat{f}_S (\cdot) \, K_\{\mathbf{x}_i\} (\cdot)\rangle^2 \right] \leq \mathbb{E}_S \left[  ||  \hat{f}_\{S_i\} - \hat{f}_S||_\mathcal{H}^2 \times \kappa \right]$. We can plugin the results from Lemma 11 to obtain similar qualitative and quantitative (up to constant factors) results as before. We will add this to our update of the paper.
>
> 2. You are absolutely correct that shifting between 'minimum' and 'minimal' can be confusing. We will make the phrasing consistent when we update the paper.
>
> 3. We assume that the kernel is uniformly bounded, that is, $K(\mathbf{x},\mathbf{x}) \leq \kappa, \forall \mathbf{x} \in X$, so our basic property in equation (7) would not depend on $\mathbf{x}$. We can expand the steps in equation 7 to make our use of the RKHS property and Cauchy-Schwarz clearer. Boundedness of the kernel is satisfied by exponential kernels, e.g. Gaussian, but also by polynomial/linear kernels assuming bounded inputs.

---

> > ### Comment · AnonReviewer2 · 2020-11-19
> > **Response**
> >
> > Thank you for your clarifications. Considering that you already analyzed the zero loss case I am now confident that you can add a small explanation for that. I do not expect much, I understand that you have very limited space, but I think it should be at least mentioned together with the explanations on the effects of it.

---

> > > ### Author Response · Authors · 2020-11-20
> > > **Thank you!**
> > >
> > > Thank you for updating your score in light of our response. We have added a remark (remark 12) to address the zero loss case, and have made a few changes to the language (minimal vs minimum, as well as remark 4) in the new version. We have also expanded on the steps in equation 7 to make that bound clearer.
> > >
> > > We hope this addresses your concerns! Thank you for your engagement with and constructive criticism of our paper.

---

### Official Review · AnonReviewer1 · 2020-10-28
**The paper establish the average leave-one-out stability bound for the interpolation solutions. The results are interesting and novel.   But I have some concerns which need authors' clarification.**

**Rating:** 6
**Confidence:** 4

**Review:**

The paper establish the average leave-one-out stability bound for the interpolation solutions, and show the above bound depends on condition number and spectral norm of kernel matrices.  The authors establishes a nice connection between numerical and statistical stability. A nice property is that among all interpolation solutions, the upper bound on stability achieves the minimum at the solution with the minimal norm. The authors then comment that the interpolation solution with minimal norm may generalize better than other interpolation solutions. The paper is clearly and well written.

Comments:

1. In Theorem 7, the authors show that the stability can be bounded by the spectral norm of $K,K^\dag$, the condition number of $K$ and the norm of $y$. It seems that this upper bound would diverge as we increase the dimension or sample size. This means that the upper bound is vacuous and may not explain the true generalization behavior of the interpolation solutions. If the upper bound is loose, then even if the interpolation solution with the minimal norm achieves the minimal upper bound, this may not convincingly show that it outperforms other interpolation solutions.

2. I have doubts on eq (10). I think the left-hand side and right-hand side of the third identity differ by the term $2K^\dag Kv_i$. If $K^\dag Kv_i\neq 0$, then this identity would not hold. Since $v_i$ can be any vector, the deduction is not convincing. I would suggest the authors to take a close look at it.

3. Lemma 5 is standard in the literature. I would suggest the authors to indicate its connection with existing results, e.g., Lemma 11 in "Learnability, Stability and Uniform Convergence"

---

> ### Author Response · Authors · 2020-11-18
> **Response to AnonReviewer1**
>
> Thank you for your comments and constructive criticism of our work. We have responded to your comments below:
>
>
> 1. Theorem 7 is for interpolating solutions. It is well known that for fixed $d$ the regularized solution has a condition number ${||K+n \lambda I||}{||(K+n \lambda I)^{-1}||}$ which is controlled by the regularization parameter $\lambda$. Consider the linear case ($K$ then is the dot product kernel). If $\lambda=0$, as in our paper, the condition number does indeed diverge for $n \to \infty$. Thus, we agree with you that if $d$ is held constant and $n$ is allowed to grow unbounded, our upper bound would also diverge and thus turn out to be vacuous (we will make this clearer in our updated draft of the paper). However, as soon as $n>d$ the solution is not an interpolating solution any longer. In fact Theorem 7 is meant to hold in the regime where $\frac{d}{n} \rightarrow \gamma > 1$ as $d,n \rightarrow \infty$. In this regime, the spectral norm and condition number of $\mathbf{K}$ and the norm of $\mathbf{K^\dagger y}$ do not diverge. While our results do not require any distributional assumptions, we can see how the bounds do not diverge under the assumption of random data. In the linear case, under appropriate distributional assumptions, we have $||\mathbf{X}^\dagger||\times ||\mathbf{y}|| \approx \frac{\sqrt{n}}{\sqrt{d} - \sqrt{n}} \rightarrow \frac{1}{\sqrt{\gamma}-1}$. For random data, the extension to the kernel case also holds since by the results of El Karoui [EK], an inner product/RBF kernel can be approximated by its linear term. We will include the above discussion in an updated draft of the paper. Reviewer 4 also raises this issue and we have responded to them in a similar fashion.
>
> 2. We apologize for the typo in equation (10), we meant to have $((\mathbf{K}_\{S_i\})^\dagger\mathbf{K}_\{S_i\} - \mathbf{K}^\dagger \mathbf{K})\mathbf{v}_i$ instead of its negative $(\mathbf{K}^\dagger \mathbf{K} - (\mathbf{K}_\{S_i\})^\dagger\mathbf{K}_\{S_i\})\mathbf{v}_i$, which is currently in place. We essentially add and subtract the term $\mathbf{K}^\dagger \mathbf{K}\mathbf{v}_i$ to go from one step to the next. This typo does not affect any of the downstream calculations. We will correct this typo in an updated draft of the paper
>
> 3. We agree with your assessment and will add this reference in an updated draft of the paper.
>
>
> [EK] El Karoui, N. (2010). The spectrum of kernel random matrices. The Annals of Statistics, 38(1), 1-50.

---

### Official Review · AnonReviewer3 · 2020-10-29
**This paper investigates kernel ridge-less regression from a stability viewpoint.**

**Rating:** 8
**Confidence:** 5

**Review:**

This paper investigates kernel ridge-less regression from a stability viewpoint by deriving its risk bounds. Using stability arguments to derive risk bounds have been widely adopting in machine learning. However, related studies on kernel ridge-less regression are still sparse. The present study fills this gap, which, in my opinion, is also one of the main contributions of the present study.

Pros:
1.  As mentioned above, this study presents some novel research into kernel ridge-less regression from a stability viewpoint.

2. The study presented here brings some novel insights into the relationship between minimizing the norm of the ERM solution and minimizing a bound on stability and also reveals the role that the condition number of the kernel matrix plays in kernel ridge-less regression, see formula (6).

3.  The paper is well presented and well polished. The analysis conducted in this paper seems to be sound.

Just one minor concern: what would happen if the boundedness assumption on the output variable, which excludes the most common Gaussian noise, is not imposed? I understand that this condition is common in learning theory but are expecting more comments.

---

> ### Author Response · Authors · 2020-11-18
> **Response to AnonReviewer3**
>
> Thank you for your positive feedback. The primary role of the boundedness of the output variable $y$ is in Lemma 10 where we use the bounded loss to obtain a bound on the stability. Since our bounds also depend on $|y|$, they would become arbitrarily bad if the output variable is unbounded. As you note, the boundedness assumption is a common one in learning theory. One possible route to addressing this case is by showing that the probability of the output variable being unbounded is low (which is true in the case of Gaussian noise), and obtaining similar bounds as in our paper with high probability.

---

### Author Response · Authors · 2020-11-20
**Paper Revision**

We have uploaded a new version of the paper that takes into account the comments from all reviewers. The significant changes are Remark 12, added in section 4, as well as an expanded discussion in the second paragraph of section 5. We have also added a reference before Lemma 5, and corrected some language and typos pointed out by the reviewers.

Thanks to all the reviewers for their constructive comments.

---

### Decision · Program_Chairs · 2021-01-07
**Final Decision**

**Decision:**

Reject

**Comment:**

The paper investigates the average stability of kernel minimal norm interpolating predictors. The main result
establishes an upper bound on a particular notion of average stability for which it is well-known that it
can be used to bound the generalization error. This upper bound holds for all interpolating predictors
from the RKHS, but it is minimized by the minimal norm predictor.

While at first glance this result looks highly interesting, a closer look reveals that the significance of the results
crucially depends on the quality of the derived upper bound. Here two reviewers raised concerns, since it is
by no means clear that even the optimized upper bound produces meaningful bounds on the generalization
performance. The authors tried to address these concerns in their response and promised to update their
paper accordingly. As a result, they added a paragraph on page 8. Unfortunately, this paragraph remains extremely
vague, in particular if it comes to the more interesting case of non-linear kernels. Here, the authors briefly refer to
a paper by El Karoui but no details are given. However, looking at El Karoui's paper it is anything but obvious whether
the results of that paper lead to reasonable upper bounds on the average stability for a sufficiently general class
of distributions.
As a result, I view the paper under review to be premature since it remains unclear if the observed optimality of the minimal norm solution is a real feature or just an artifact due to an upper bound that is simply too loose to make any conclusion.